# Dual leucine zipper kinase is required for mechanical allodynia and microgliosis after nerve injury

Josette J Wlaschin[1†], Jacob M Gluski[1†], Eileen Nguyen[2†], Hanna Silberberg[1], James H Thompson[2], Alexander T Chesler[2*], Claire E Le Pichon[1*]

[1]Eunice Kennedy Shriver National Institute of Child Health and Human Development, National Institutes of Health, Bethesda, United States; [2]National Center for Complementary and Integrative Health, National Institutes of Health, Bethesda, United States

**Abstract** Neuropathic pain resulting from nerve injury can become persistent and difficult to treat but the molecular signaling responsible for its development remains poorly described. Here, we identify the neuronal stress sensor dual leucine zipper kinase (DLK; *Map3k12*) as a key molecule controlling the maladaptive pathways that lead to pain following injury. Genetic or pharmacological inhibition of DLK reduces mechanical hypersensitivity in a mouse model of neuropathic pain. Furthermore, DLK inhibition also prevents the spinal cord microgliosis that results from nerve injury and arises distant from the injury site. These striking phenotypes result from the control by DLK of a transcriptional program in somatosensory neurons regulating the expression of numerous genes implicated in pain pathogenesis, including the immune gene *Csf1*. Thus, activation of DLK is an early event, or even the master regulator, controlling a wide variety of pathways downstream of nerve injury that ultimately lead to chronic pain.
DOI: https://doi.org/10.7554/eLife.33910.001

**\*For correspondence:**
alexander.chesler@nih.gov (ATC);
claire.lepichon@nih.gov (CELP)

[†]These authors contributed equally to this work

**Competing interests:** The authors declare that no competing interests exist.

## Introduction

Neuropathic pain is a disease of the somatosensory system that is characterized by heightened sensitivity to innocuous and noxious stimuli (allodynia and hyperalgesia). This maladaptation of the pain detection system results in an important evolutionary warning signal (for example guarding a wound during tissue healing) persisting and becoming chronic and debilitating (*Ji and Strichartz, 2004*; *Basbaum et al., 2009*; *von Hehn et al., 2012*). Despite recent advances, the mechanisms underlying the development of neuropathic pain are still not fully understood. Characterization of signaling pathways immediately downstream of nerve injury could thus help understand the molecular and cellular changes that are critical for its development and uncover new targets for the treatment of chronic pain.

Peripheral nerve injury is a common cause of neuropathic pain in humans; the spared nerve injury (SNI) rodent model reliably replicates this type of chronic pain. In this model, partial sciatic nerve axotomy robustly results in the development of mechanical allodynia that can be measured using a reflexive behavior phenotype (paw withdrawal to a punctate touch stimulus) (*Decosterd and Woolf, 2000*; *Bourquin et al., 2006*). A correlate of the mechanical allodynia that arises following SNI is a neuroinflammatory response in the spinal cord, distant from the injury site, and characterized by an intense microgliosis in the dorsal horn of the spinal cord in the vicinity of the central nerve terminals of the injured DRG (dorsal root ganglion) neurons (*Svensson et al., 1993*; *Ji et al., 2013*). This gliosis has been the focus of many recent pain studies as the microglia are believed to contribute to an

alteration in the pain transmission circuitry in the spinal cord dorsal horn (*Beggs et al., 2012*; *Clark and Malcangio, 2014*; *Grace et al., 2014*; *Guan et al., 2016*).

Dual leucine zipper kinase (DLK; *Map3k12*) is the critical initiator of a neuronal stress response to acute nerve injury in many neuron types across a wide range of phyla (*Yan et al., 2009*; *Hammarlund et al., 2009*; *Miller et al., 2009*; *Watkins et al., 2013*; *Le Pichon et al., 2017*). DLK signaling plays an important role in determining outcomes for the injured neurons including axon regeneration and neuronal survival (*Yan et al., 2009*; *Hammarlund et al., 2009*; *Xiong et al., 2010*; *Xiong and Collins, 2012*; *Watkins et al., 2013*) as well as axon degeneration and neuronal death (*Miller et al., 2009*; *Ghosh et al., 2011*; *Watkins et al., 2013*; *Welsbie et al., 2013*; *Simon et al., 2016*). However, to date, a role for DLK in the development of neuropathic pain has not been examined.

In this study, we show that genetic deletion or pharmacological inhibition of DLK greatly reduces the development of mechanical allodynia as well as the spinal cord microgliosis resulting from spared nerve injury. We find that DLK acts as a 'master regulator' of transcriptional changes responsible for the establishment of neuropathic pain phenotypes caused by nerve injury. These include the upregulation of immune gene *Csf1* (colony stimulating factor 1) in primary sensory neurons. We thus describe for the first time the DLK-dependence of the neuronally-derived immune signal (Csf1) responsible for initiating a microglial response at a site distant from the injury (*Guan et al., 2016*). Our results demonstrate an essential role of DLK in the establishment of nerve injury-induced changes, including neuro-immune signaling, that result in neuropathic pain sensitization. By shedding light on the mechanisms of establishment of pain, this work can pave the way towards new treatments.

## Results

### DLK is required for mechanical allodynia in the SNI model of neuropathic pain

Spared nerve injury results in long lasting hypersensitivity to touch stimuli. To test whether DLK function was important in the establishment of mechanical allodynia, we conditionally deleted DLK (*Map3k12*) expression in adult mice by using an inducible strategy that avoids developmental effects (*Map3k12*[fl/fl] crossed to CAG-CreER[+/-]; [*Pozniak et al., 2013*]). We assessed the behavioral effect of SNI in these DLK cKO (conditional DLK knockout) versus control mice (*Map3k12*[fl/fl]; CAG-CreER[-/-], referred to as DLK WT) by measuring paw withdrawal thresholds to calibrated von Frey filaments. Strikingly, DLK cKO mice were protected from developing static mechanical allodynia to punctate stimulation of their ipsilateral hind paws as compared with DLK WT littermates (*Figure 1A–B*). Moreover, the protection was long lasting and persisted for at least 4 weeks following injury. DLK cKO mice also showed less dynamic mechanical hypersensitivity to a light brushing stimulus (*Figure 1—figure supplement 1*). Importantly, deleting DLK results in no measureable behavioral deficits in the absence of injury, including normal baseline von Frey thresholds (*Figure 1A*), and normal motor and cognitive behaviors as shown in previous studies (*Pozniak et al., 2013*; *Le Pichon et al., 2017*). To examine how early the DLK cKO benefit was present, we tested more acute time points following SNI injury (1, 3, 5, and 7 days post injury or dpi) in a separate cohort of mice. DLK cKO mice showed less hypersensitivity as early as one dpi (*Figure 1—figure supplement 2*). Thus, in the absence of DLK signaling, acute and long-lasting allodynia is blocked.

### DLK activity controls the expression of many pain-related genes

To better understand the mechanisms of these protective effects, we investigated the molecular consequences downstream of DLK signaling in peripheral sensory neurons. DLK is upstream of a kinase signaling cascade resulting in a transcriptional response to injury via phosphorylation of JNK (c-Jun N-terminal kinase) and the transcription factor c-Jun (*Watkins et al., 2013*; *Ghosh et al., 2011*; *Fan et al., 1996*; *Cavalli et al., 2005*; *Fernandes et al., 2014*; *Larhammar et al., 2017*). Given the central role of DLK in transcriptional regulation, we reasoned that this kinase might be important for regulation of multiple genes involved in neuropathic pain.

We first examined the effect of DLK deletion on the induction of transcription factors phospho-c-Jun (p-c-Jun) and ATF3 under two conditions known to cause neuropathic pain, SNI and intraplantar

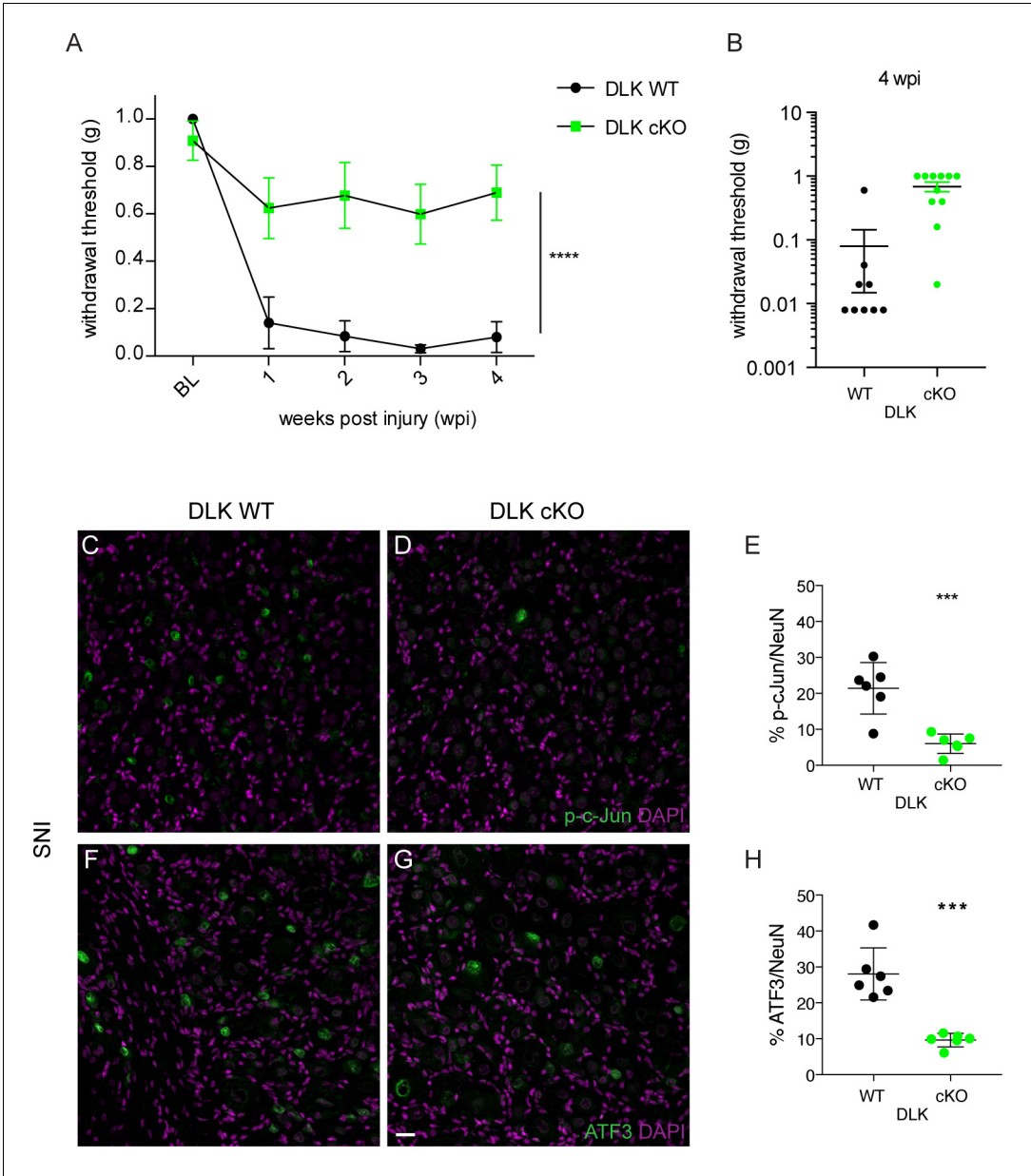

**Figure 1.** DLK is necessary for the development of mechanical allodynia and induction of injury markers after spared nerve injury (SNI). (**A**) DLK deletion is protective in the von Frey behavior assay following SNI at 1, 2, 3 and 4 weeks post injury (wpi). Baseline (BL) mechanical sensitivity thresholds are comparable between DLK WT (n = 9) and DLK cKO (n = 11). ****p<0.0001 by repeated measures ANOVA. (**B**) Scatter graph to illustrate individual points for DLK WT and DLK cKO at 4 weeks post injury. (**C–H**) Deletion of DLK reduces injury-induced transcription factors p-c-Jun and ATF3 in DRG following SNI. Representative images of ipsilateral L4 DRG 1 week after SNI stained for p-c-Jun in DLK WT (**C**) and DLK cKO (**D**) quantified in (**E**), and for ATF3 in DLK WT (**F**) and DLK cKO (**G**) quantified in (**H**). n = 6 per genotype. Scale bar in (**G**) valid for all images in this figure: 20 μm. ***p<0.001 by 2-tailed Student's t test.

DOI: https://doi.org/10.7554/eLife.33910.002

The following source data and figure supplements are available for figure 1:

**Source data 1.** data plotted in *Figure 1A*; von Frey behavior testing in DLK WT and cKO mice.
DOI: https://doi.org/10.7554/eLife.33910.006

**Source data 2.** data plotted in *Figure 1E*; p-cJun-positive nuclear counts in ipsilateral L4 DRG 7 days after SNI in DLK WT vs cKO mice.
DOI: https://doi.org/10.7554/eLife.33910.007

*Figure 1 continued*

**Source data 3.** data plotted in *Figure 1H*; ATF3-positive nuclear counts in ipsilateral L4 DRG 7 days after SNI in DLK WT vs cKO mice.
DOI: https://doi.org/10.7554/eLife.33910.008

**Source data 4.** data plotted in *Figure 1—figure supplement 1*; dynamic brush scores 7 weeks after SNI in DLK WT vs cKO mice.
DOI: https://doi.org/10.7554/eLife.33910.009

**Source data 5.** data plotted in *Figure 1—figure supplement 2*; von Frey testing in DLK WT vs cKO mice at 1, 3, 5 and 7 days post SNI.
DOI: https://doi.org/10.7554/eLife.33910.010

**Source data 6.** data plotted in *Figure 1E*; p-cJun-positive nuclear counts in ipsilateral L4 DRG 48 hr after hind paw formalin injection in DLK WT vs cKO mice.
DOI: https://doi.org/10.7554/eLife.33910.011

**Source data 7.** data plotted in *Figure 1E*; ATF3-positive nuclear counts in ipsilateral L4 DRG 48 hr after hind paw formalin injection in DLK WT vs cKO mice.
DOI: https://doi.org/10.7554/eLife.33910.012

**Figure supplement 1.** DLK deletion prevents allodynia to dynamic mechanical stimulus after spared nerve injury (SNI).
DOI: https://doi.org/10.7554/eLife.33910.003

**Figure supplement 2.** DLK is necessary for the development of mechanical allodynia after spared nerve injury (SNI).
DOI: https://doi.org/10.7554/eLife.33910.004

**Figure supplement 3.** DLK deletion reduces injury markers in DRG in the formalin model of neuropathic pain.
DOI: https://doi.org/10.7554/eLife.33910.005

formalin injection. Importantly, these procedures had previously been shown to result in the upregulation of both markers (*Zhuang et al., 2006*; *Bráz and Basbaum, 2010*). Both p-c-Jun and ATF3 were induced in injured DRGs from DLK WT mice but this was prevented in DLK cKO mice under both conditions (by 65–71% with SNI, *Figure 1C–H*; by 90–95% in the formalin model, *Figure 1—figure supplement 3*). Therefore, both p-c-Jun and Atf3 can be considered downstream markers of DLK signaling in injured DRG neurons. This is consistent with previous reports of injury-induced ATF3 as downstream of DLK in another neuron type (retinal ganglion cells; (*Watkins et al., 2013*; *Fernandes et al., 2013*).

We next examined the potential DLK-dependence of changes in expression levels of genes induced following nerve injury that correlate with increased pain sensations. We therefore referred to a published transcriptomic dataset containing the list of genes whose expression is altered in the DRG following sciatic nerve transection (sample GSM1626431 from series GSE66619 [*Guan et al., 2016*]). Among these genes, we examined 10 among the top 25 whose expression is altered 7 days after sciatic nerve transection (*Table 1*). Some but not all these genes were present on a curated list of DLK-dependent genes from a microarray dataset of retinas 3 days following optic nerve crush (*Watkins et al., 2013*).

We measured mRNA expression levels at 7 dpi using quantitative in situ hybridization (RNAscope) of *Atf3* as well as two neuropeptides previously documented to increase after SNI, galanin (*Gal* [*Kashiba et al., 1992*]) and neuropeptide Y (*Npy* [*Ma and Bisby, 1998*]). Strikingly, we found DLK to be responsible for injury-induced transcriptional increases of all three genes. In wild type mice, SNI upregulates *Atf3*, *Gal* and *Npy*, (*Figure 2A,D,G*). By contrast, SNI fails to upregulate these genes in DLK cKO littermates (*Figure 2B–C,E–F,H–I*). Other injury-induced genes that we confirmed were DLK-dependent in DRG neurons included *Cckbr*, *Ecel1*, *Gpr151*, *Nts*, *Sox11*, and *Sprr1a* (*Figure 2—figure supplement 1*). In all cases, the increased expression in response to injury was significantly attenuated in DLK cKO mice (*Figure 2—figure supplement 1A–R*).

We next examined whether DLK might also be important for genes that are downregulated after injury as well. A recent study reported that SNI-induced reduction of Serine protease inhibitor 3 (*Serpina3n*) expression enhances mechanical allodynia (*Vicuña et al., 2015*). We found *Serpina3n* levels to be negatively regulated by DLK as evidenced by significantly higher mRNA levels after SNI in DLK cKO mice than those seen in DLK WT mice (*Figure 2J–L*). Therefore, the injury-induced expression

**Table 1.** Top 25 genes with altered expression 7 days following sciatic nerve transection from (*Guan et al., 2016*)

Genes are sorted by significance, then by fold change. The genes we examined for DLK-dependence by in situ hybridization are indicated by an asterisk *

|  | Gene | log2 fold_change |
|---|---|---|
| 1 | Sprr1a * | 6.96879 |
| 2 | Npy * | 5.89862 |
| 3 | Gpr151 * | 5.72999 |
| 4 | Cckbr * | 5.35581 |
| 5 | Ecel1 * | 4.97284 |
| 6 | Atf3 * | 4.95333 |
| 7 | Gal * | 4.77653 |
| 8 | Nts * | 4.60215 |
| 9 | Hrk | 4.55133 |
| 10 | Fst | 4.09785 |
| 11 | Lmo7 | 3.76662 |
| 12 | Car3 | 3.48156 |
| 13 | Sox11 * | 3.37483 |
| 14 | Sema6a | 3.32589 |
| 15 | Mmp16 | 3.31338 |
| 16 | Flrt3 | 3.28661 |
| 17 | Loxl2 | 3.25419 |
| 18 | Flnc | 3.21128 |
| 19 | Sez6l | 3.20287 |
| 20 | Adam8 | 2.98222 |
| 21 | Stmn4 | 2.89975 |
| 22 | Xdh | 2.77339 |
| 23 | Gadd45a | 2.66746 |
| 24 | Sdc1 | 2.59248 |
| 25 | Csf1 * | 2.51895 |

DOI: https://doi.org/10.7554/eLife.33910.013

changes of at least 10 genes in the DRG after SNI are DLK-dependent, suggesting that DLK is a regulator of numerous pain-associated genes.

One additional gene we focused on was colony stimulating factor 1 (*Csf1*), due to a recent report that this cytokine becomes neuronally expressed upon nerve injury, and is responsible for the induction of spinal cord microgliosis following nerve injury (*Guan et al., 2016*). The authors of this work also showed that Csf1 is required for the establishment of mechanical allodynia. We found that *Csf1* induction failed to occur in DLK cKO DRG neurons examined 7 days post-SNI (*Figure 3A–C*), demonstrating that *Csf1* upregulation depends on DLK signaling, and prompting us to examine whether microgliosis was affected in mice lacking DLK.

## DLK is necessary for SNI-induced spinal cord microgliosis

Microglia have been implicated as important contributors to the development of mechanical allodynia by altering the pain transmission circuitry in the spinal cord dorsal horn (*Beggs et al., 2012*; *Clark and Malcangio, 2014*; *Grace et al., 2014*; *Guan et al., 2016*). Because we had found DLK to control Csf1 expression, we examined the dorsal horn microgliosis that occurs following SNI in mice lacking DLK. Microglia in the whole lumbar spinal cord were visualized at 8 days post injury using Iba1 (ionized calcium binding adapter molecule 1) immunostaining and tissue clearing (iDisco

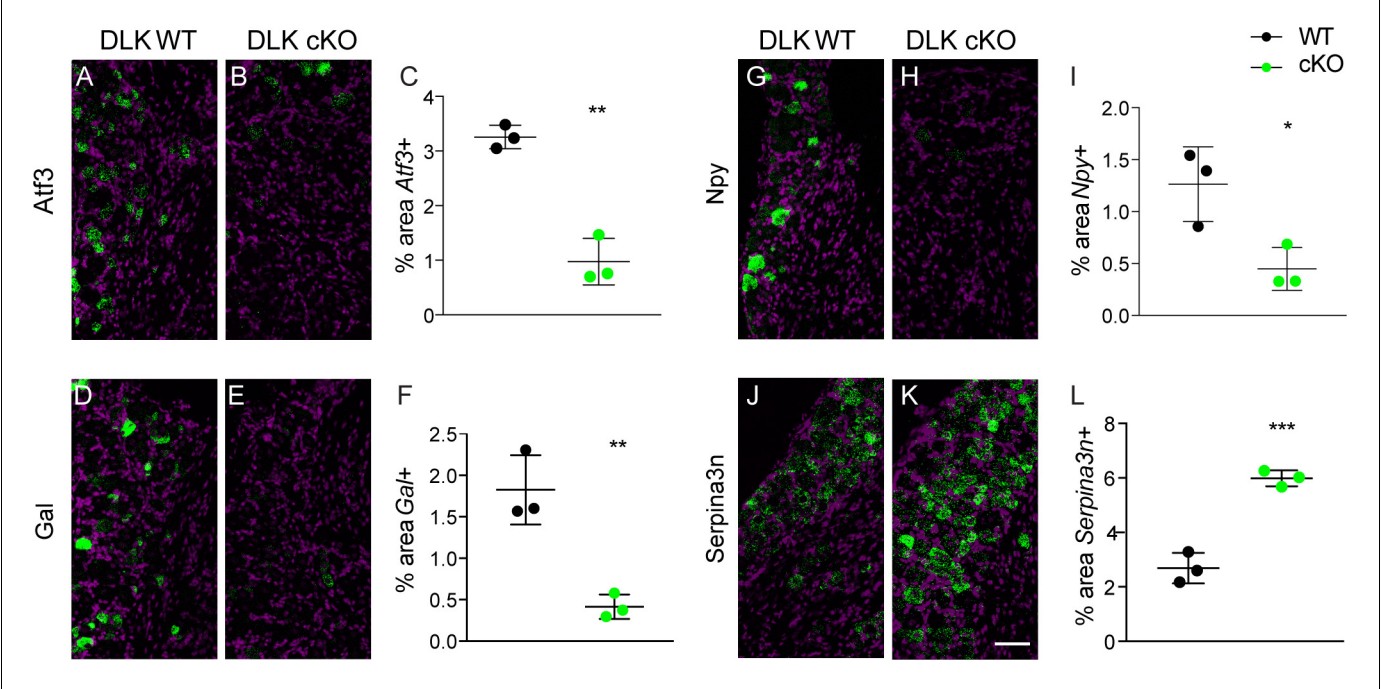

**Figure 2.** DLK is upstream of injury-associated genesimplicated in the establishment of neuropathic pain: Atf3, Gal, Npy, and Serpina3n. Representative images and quantification of expression of 4 genes whose expression changes following sciatic nerve axotomy: *Atf3* (A–C), *Gal* (D–F), *Npy* (G–I), and *Serpina3n* (J–L) by in situ hybridization in DLK WT vs DLK cKO ipsilateral L3 DRG 1 week after SNI (7 dpi). Scale bar 100 μm. *p<0.05, **p<0.01, ***p<0.001 by 2-tailed Student's t test. Averaged data for n = 4–6 sections quantified in n = 3 male mice per genotype.
DOI: https://doi.org/10.7554/eLife.33910.014

The following source data and figure supplement are available for figure 2:

**Source data 1.** data plotted in *Figure 2*; quantification of in situ hybridization for *Atf3, Gal, Npy* and *Serpina3n* in DLK WT vs cKO in ipsilateral L4 DRG 7 days post SNI.
DOI: https://doi.org/10.7554/eLife.33910.016

**Source data 2.** data plotted in *Figure 2—figure supplement 1*; quantification of in situ hybridization for *Cckbr, Ecel1, Gpr151, Nts, Sox11*, and *Sprr1a* in DLK WT vs cKO in ipsilateral L4 DRG 7 days post SNI.
DOI: https://doi.org/10.7554/eLife.33910.017

**Figure supplement 1.** DLK is upstream of multiple genes implicated in the establishment of neuropathic pain.
DOI: https://doi.org/10.7554/eLife.33910.015

(*Renier et al., 2014*); *Figure 3D*; see also *Video 1*). Strikingly, we found that microglial recruitment following SNI requires intact DLK signaling. SNI-induced ipsilateral dorsal horn microgliosis was prevented in mice lacking DLK compared to wild type controls (*Figure 3D–H*), while there was no difference between genotypes in the microglial density in contralateral dorsal spinal cord (*Figure 3I*). Therefore, DLK is essential for the spinal cord microgliosis that occurs at the central nerve terminals of DRG neurons following peripheral nerve injury, at a considerable distance from the injury site.

## Acute time course of DLK-dependent gene expression changes and spinal cord microgliosis following sciatic nerve injury

To gain an understanding of the acute expression time course for a subset of genes transcriptionally regulated by DLK (*Atf3, Csf1, Gal*), we used whole sciatic nerve transection (SNT) since this injury results in the vast majority of L4 DRG neurons (87% ± 3.44 (SEM), n = 5) expressing *Atf3* in the L4 DRG by 48 hr post injury. We found that injury-induced neuronal *Atf3* expression plateaued early: levels were already significantly increased in ipsilateral versus contralateral L4 DRG by 18 hr (*Figure 4A–B*). The onset of injury-induced neuronal *Csf1* expression lagged behind *Atf3* with a sharp increase between 18 and 24 hr (*Figure 4A,C*). Ipsilateral *Gal* expression was significantly increased compared to contralateral by 48 hr (*Figure 4D–E*).

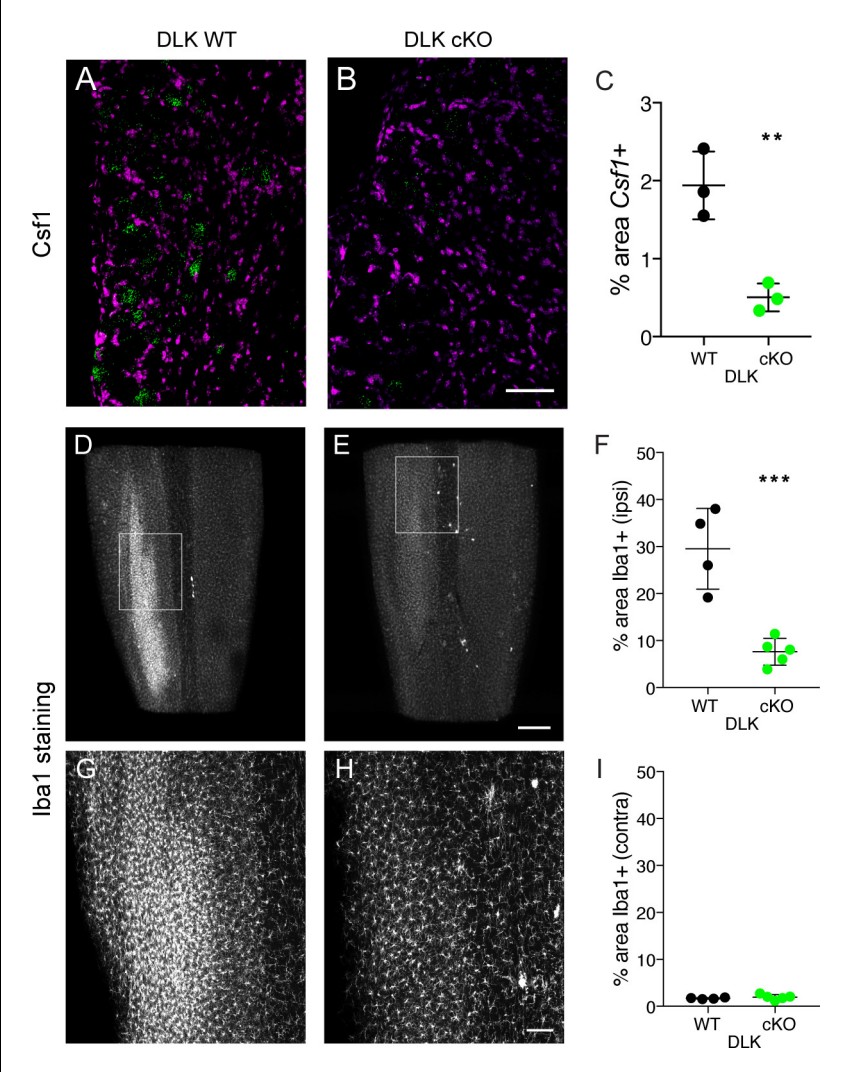

**Figure 3.** DLK is necessary for Csf1 upregulation in DRG neurons and for spinal cord microgliosis elicited by SNI. (**A–C**) Representative images and quantification of *Csf1* gene expression changes following SNI by in situ hybridization in DLK WT vs DLK cKO L3 DRG 7 days post SNI. Scale bar 100 µm. **p<0.01, by 2-tailed Student's t test. Averaged data for n = 4–6 sections quantified in n = 3 mice per genotype. (**D–I**) Representative images and quantification of microgliosis in DLK WT vs DLK cKO spinal cords 8 days post-SNI. Iba1 staining in cleared spinal cords (top view, longitudinal plane) showing the SNI-induced microgliosis in the ipsilateral (left) dorsal horn of DLK WT spinal cord (**D, G**) that does not occur in the DLK cKO (**E, H**). (**G**) and (**H**) are higher magnification images of boxed areas in (**D**) and (**E**) respectively. Quantification of dorsal horn microgliosis in ipsilateral (**F**) and contralateral (**I**) spinal cord of DLK WT (n = 4) vs DLK cKO (n = 5). Scale bar in (**D, E**) 400 µm. Scale bar in (**G, H**) 100 µm. ***p=0.0010 by 2-tailed Student's t test.

DOI: https://doi.org/10.7554/eLife.33910.018

The following source data is available for figure 3:

**Source data 1.** data plotted in *Figure 3C*; quantification of in situ hybridization for *Csf1* in DLK WT vs cKO in ipsilateral L4 DRG 7 days post SNI.
DOI: https://doi.org/10.7554/eLife.33910.019

**Source data 2.** data plotted in *Figure 3F*; quantification of Iba1-positive signal in DLK WT vs cKO dorsal spinal cord ipsilateral to injury at 8 dpi.
DOI: https://doi.org/10.7554/eLife.33910.020

**Source data 3.** data plotted in *Figure 3I*; quantification of Iba1-positive signal in DLK WT vs cKO dorsal spinal cord contralateral to injury at 8 dpi.
DOI: https://doi.org/10.7554/eLife.33910.021

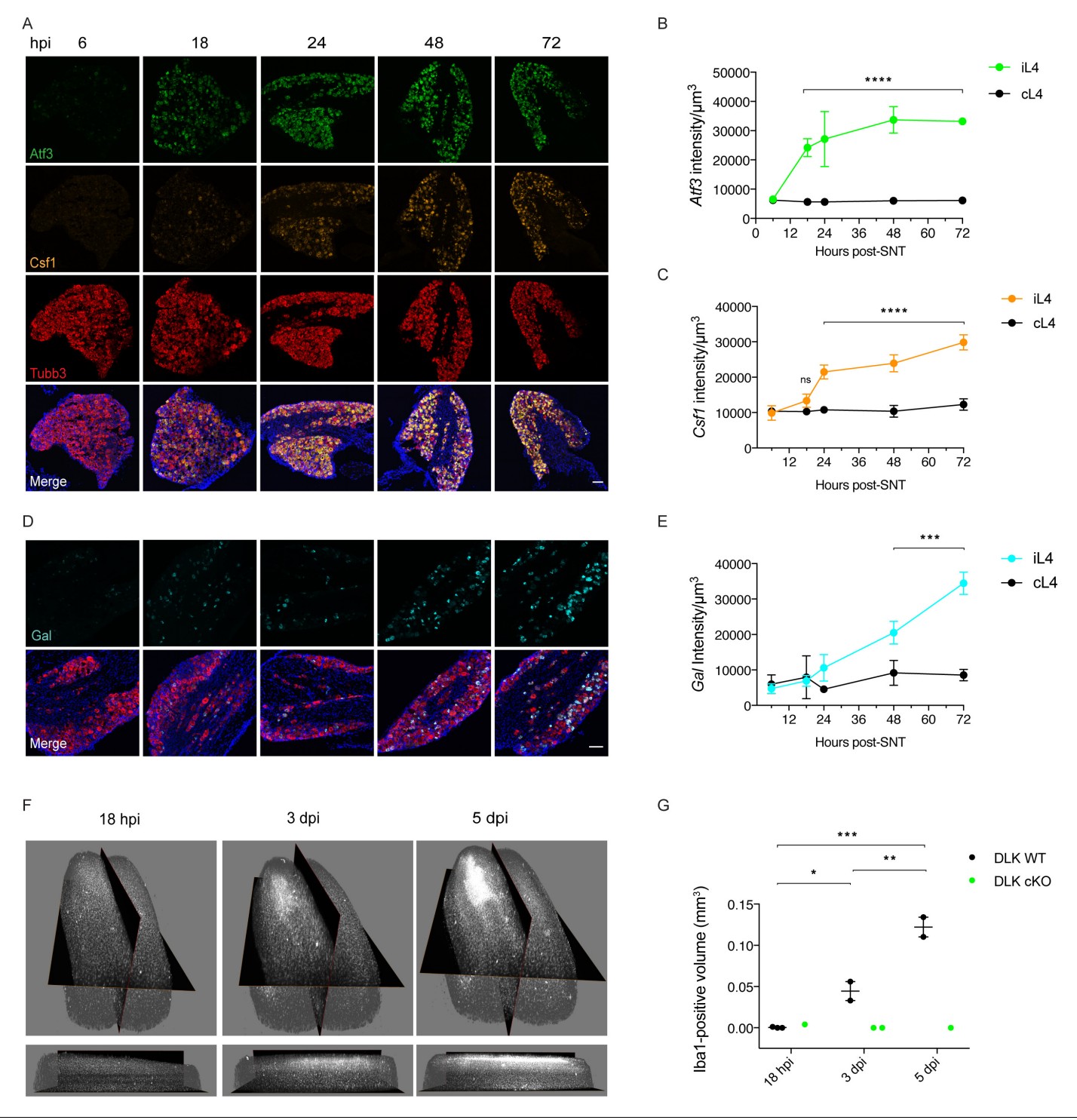

**Figure 4.** Time course of DLK-dependent gene expression in DRG neurons and corresponding spinal cord microgliosis. (A–C) Representative images and quantification of gene expression changes in ipsilateral L4 DRGs of WT mice 6, 18, 24, 48 and 72 hr post-SNT (sciatic nerve transection). (A) and (D) Representative images of *Atf3* (green), *Csf1* (orange), *Gal* (cyan) and *Tubb3* (red) in situ hybridizations in iL4 DRGs after SNT. Scale bars 100 μm. Quantification of *Atf3* (B), *Csf1* (C), and *Gal* (E) expression changes in L4 DRGs ipsilateral (green, orange, cyan iL4) vs contralateral (black, cL4) to injury 6–72 hr post-SNT. n = 4 (2 males, 2 females) for 6, 18, 24, and 48 hr groups, n = 2 (1 male, 1 female) for 72 hr group. ns: not significant; ***p<0.001, ****p<0.0001 by 2-way ANOVA followed by Sidak's multiple comparisons post test comparing values between ipsilateral and contralateral L4 DRG at each time point. Note in (D), in representative images at 48 and 72 hr, that *Gal* comes on in fewer cells than for *Atf3* and *Csf1* (A), and these include a mixture of large and small diameter neurons. (F–G) Iba1 immunolabeling in cleared lumbar spinal cord shows no region of microglial reactivity at 18 hr

*Figure 4 continued on next page*

*Figure 4 continued*

post-SNI. Microgliosis becomes evident by 3 dpi and robust by 5 dpi. Representative images of 3D views from the top and the ipsilateral side (**F**) and quantification (**G**). DLK cKO prevents microgliosis in lumbar spinal cord 3 and 5 days post-SNI, as observed previously at 7 dpi (*Figure 3D–H*). *p<0.05, **p<0.01, ***p<0.001 by one-way ANOVA followed by Tukey's multiple comparisons.

DOI: https://doi.org/10.7554/eLife.33910.023

The following source data is available for figure 4:

**Source data 1.** data plotted in *Figure 4B,C and E*; quantification of time course of expression of *Atf3*, *Csf1* and *Gal* in ipsilateral L4 DRGs normalized by total neuronal volume sampled as labeled by *Tubb3*.

DOI: https://doi.org/10.7554/eLife.33910.024

**Source data 2.** data plotted in *Figure 4G*; quantification of Iba1-positive volume in dorsal spinal cord ipsilateral to injury at the indicated time points.

DOI: https://doi.org/10.7554/eLife.33910.025

We carried out a parallel spared nerve injury time course to examine the onset of spinal cord microgliosis following injury. The appearance of the microglial 'cloud' in the spinal cord ipsilateral to injury lags behind the increase in *Csf1* expression in the DRG, in line with the report showing that *Csf1* expression by injured neurons is required for the microgliosis (*Guan et al., 2016*). It is appreciable by 3 days, strong by 5 days post injury, but not yet apparent by 18 hr post injury (*Figure 4F–G*). Our time course results are consistent with previous descriptions of microgliosis resulting from traumatic nerve injury (*Guan et al., 2016*). The immunolabeling and clearing of whole spinal cords allows a visualization of the injury-induced microglial cloud in its entirety (see *Video 1*).

## Treatment with DLK inhibitor GNE-3511 prevents injury-induced transcriptional changes, mechanical allodynia and microgliosis

Knowing the time course of onset of injury-induced transcriptional changes and microgliosis, we next tested whether DLK inhibition would be sufficient to prevent mechanical allodynia and microgliosis after an injury had already occurred. We dosed wild type mice twice daily with DLK inhibitor compound GNE-3511 (*Patel et al., 2015*) at 75 mg/kg, starting 16–18 hr post SNI (*Figure 5D*). This time point coincides with high *Atf3* levels but precedes a sharp increase in *Csf1* expression (*Figure 4A,C*) and the establishment of microgliosis (*Figure 4F–G*). Dose level and frequency were selected based on prior pharmacokinetic and pharmacodynamic studies (*Patel et al., 2015*; *Le Pichon et al., 2017*). To first verify the inhibitor compound indeed reduced DLK signaling in DRG neurons, we examined gene expression of *Atf3* and *Csf1* at 5 dpi in L4 DRGs in GNE-3511 versus vehicle-treated mice. GNE-3511 reversed transcriptional induction of Atf3 (which was already elevated at 18 hr, when dosing was initiated, *Figure 4B*) and prevented Csf1 induction (*Figure 5A–C*). We then proceeded to behavior testing and measurements of microgliosis in a separate group of animals. Strikingly, GNE-3511-mediated inhibition of DLK significantly prevented development of mechanical allodynia (*Figure 5E*), as well as the establishment of microgliosis (*Figure 5F–H*; *Videos 1–2*).

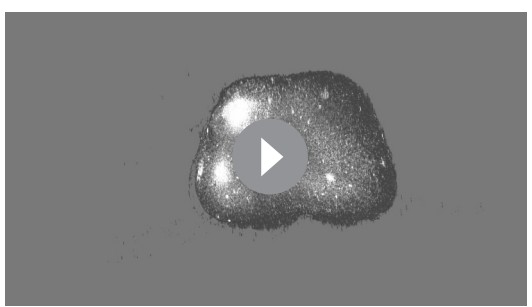

**Video 1.** untreated spinal cord, 5 dpi
DOI: https://doi.org/10.7554/eLife.33910.022

## Discussion

Our results thus reveal DLK signaling to be a critical step in the molecular pathway that results in the establishment of neuropathic pain following peripheral nerve injury. Both the genetic and pharmacological methods of DLK inhibition demonstrate that the transcriptional response downstream of DLK causes the spinal cord microgliosis as well as heightened withdrawal responses to touch stimuli.

Our findings provide a mechanistic explanation of how DLK deletion blocks the development of mechanical allodynia after SNI (*Figure 6*). Although our Cre driver line results in global DLK deletion, in this work we focused on the effect of

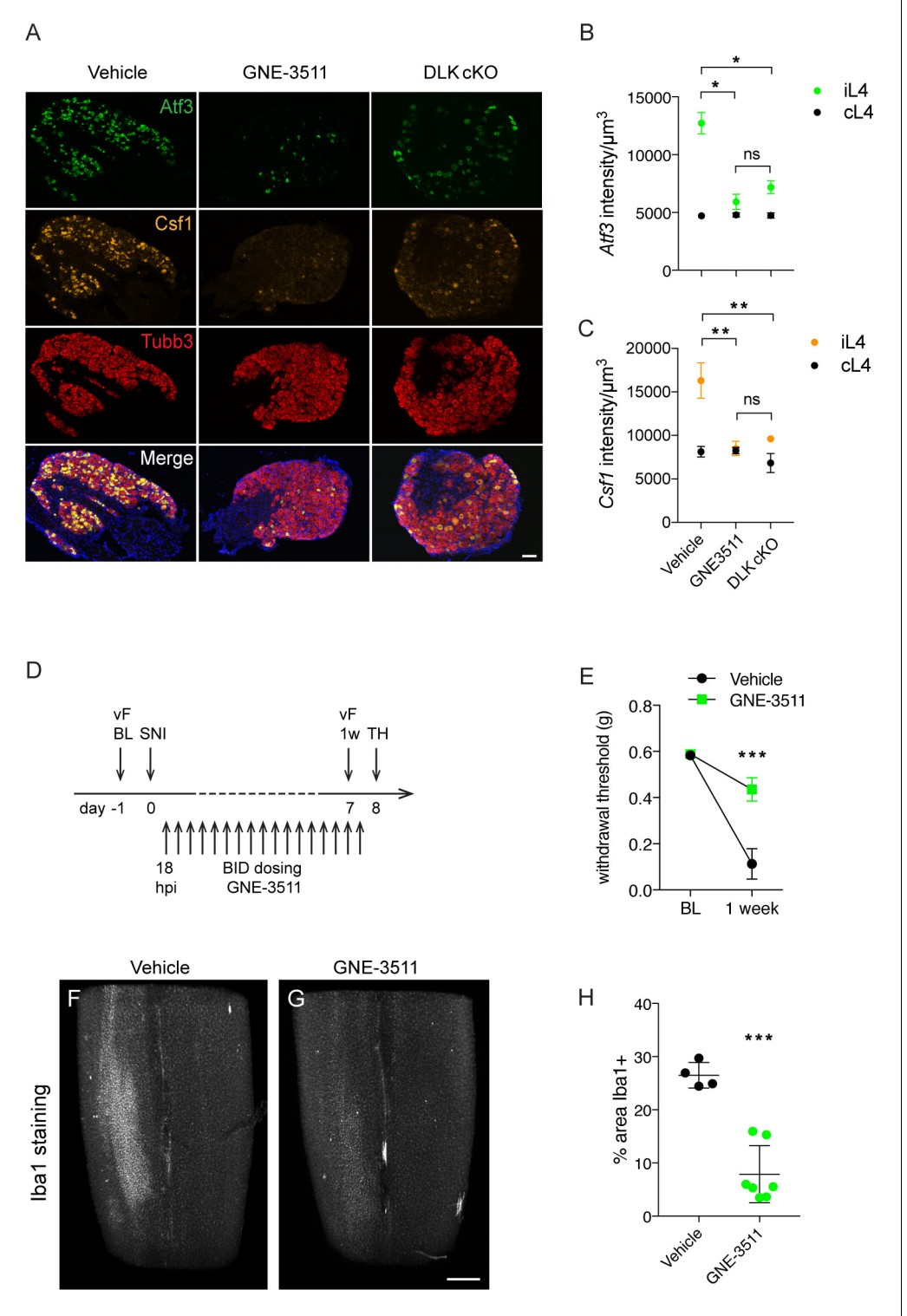

**Figure 5.** DLK inhibition prevents DLK-dependent gene upregulation, mechanical allodynia, and microgliosis after SNI. (A–C) GNE-3511 treatment prevents upregulation of DLK-dependent genes in the DRG after SNI. (A) Representative images of *Atf3* (green), *Csf1* (orange), and *Tubb3* (red) in situ hybridizations in ipsilateral L4 DRGs from vehicle-treated, GNE3511-treated, and DLK cKO mice 5 days post-SNI. Scale bar 100 µm. (B, C) Quantification of neuronal gene expression intensity in ipsilateral L4 DRGs at 5 dpi showing reduced expression of *Atf3* (B) and *Csf1* (C) in DLK cKO and GNE-3511 vs vehicle treated mice (n = 2 per group). *p<0.05, **p<0.01 by Tukey's multiple comparisons test. (D) DLK inhibitor study design. vF: von Frey; BL: baseline; SNI: spared nerve
*Figure 5 continued on next page*

*Figure 5 continued*

injury; 1 w: 1 week post SNI; TH: tissue harvest. Twice daily dosing of GNE-3511 began at 18 hr post injury (hpi). (**E**) DLK inhibitor GNE-3511 prevents mechanical allodynia measured 1 week post injury. Vehicle n = 12; GNE-3511 n = 17. ***p=0.0004 for genotype by 2-way ANOVA. (**F–H**) GNE-3511 prevents the spinal cord microgliosis elicited by SNI as shown by Iba1 staining in cleared immunostained lumbar spinal cord. Representative images from vehicle (**F**) vs GNE-3511 treated mice (**G**) and quantification of Iba1-positive signal in ipsilateral spinal cord (**H**). Vehicle n = 4; GNE-3511 n = 7. ***p=0.0001 by 2-tailed Student's t test. Scale bar in (**F**), (**G**) 400 µm.

DOI: https://doi.org/10.7554/eLife.33910.026

The following source data is available for figure 5:

**Source data 1.** data plotted in *Figure 5B,C*; quantification of in situ signal intensity at 5 dpi for *Atf3* and *Csf1* in ipsilateral and contralateral L4 DRGs normalized by total neuronal volume sampled as labeled by *Tubb3*.

DOI: https://doi.org/10.7554/eLife.33910.027

**Source data 2.** data plotted in *Figure 5E*; von Frey thresholds (>5 responses out of 10 stimulations) in vehicle vs GNE-3511-treated at 7 dpi.

DOI: https://doi.org/10.7554/eLife.33910.028

**Source data 3.** data plotted in *Figure 5H*; quantification of Iba1-positive signal in vehicle vs GNE-3511-treated dorsal spinal cord ipsilateral to injury at 8 dpi.

DOI: https://doi.org/10.7554/eLife.33910.029

DLK loss in peripheral neurons, and found many cell-autonomous gene expression changes in the injured DRGs that are DLK-dependent. It will be of great interest to delete DLK specifically in sensory neurons to assess whether this recapitulates the effect of global DLK deletion or whether DLK activity in other cell types is also important to produce the mechanical allodynia we have described.

It is particularly notable that DLK activity is essential for *Csf1* upregulation in sensory neurons. In light of recent work showing that neuronal expression of Csf1 is necessary and sufficient for the establishment of mechanical allodynia via recruitment of microglia to the dorsal horn (*Guan et al., 2016*), we now provide an explanation for how nerve injury upregulates Csf1 expression. Our work places DLK upstream of neuronal Csf1 expression, revealing a mechanism whereby an immune response can be elicited by injured neurons in the form of microglial recruitment at a site distant from the injury. The significance of this finding extends beyond the case of traumatic nerve injury to other contexts of neuronal injury and dysfunction, for example neurodegenerative disease, in which gliosis is a frequent correlate of neuronal pathology and neuroinflammation is known to play an important role (*Ransohoff, 2016*; *Salter and Stevens, 2017*). In fact, our previous work found that deleting DLK in a mouse model of neurodegenerative disease also reduced microgliosis (*Le Pichon et al., 2017*), and another study had also found that DLK deletion prevented microgliosis resulting from facial motor neuron injury (*Itoh et al., 2014*). Interestingly, a recent report found that a related injury signaling pathway, Sarm1, also controls a neuro-immune response (*Wang et al., 2018*). The authors reported that Sarm1 is responsible for upregulating several chemokine ligand genes via JNK and c-Jun, but not Csf1, whose upregulation is however c-Jun-dependent.

Neuropathic pain is poorly treated by current pain therapies; thus, finding effective drugs to prevent or treat this debilitating chronic syndrome is of critical importance especially in light of the current opioid epidemic (*Wilkerson et al., 2016*). Our identification of DLK as an upstream regulator of pathways that result in chronic pain provides a new logic for designing and testing therapeutic strategies.

## Materials and methods

**Key resources table**

| Reagent type (species) or resource | Designation | Source or reference | Identifiers | Additional information |
|---|---|---|---|---|
| Commercial assay/kit- RNAscope | RNAscope Fluorescent Multiplex Detection Reagents | Advanced Cell Diagnostics | ACD:320851 | |

*Continued on next page*

*Continued*

| Reagent type (species) or resource | Designation | Source or reference | Identifiers | Additional information |
|---|---|---|---|---|
| Commercial assay/kit- RNAscope | RNAscope Probe-Mm-Atf3; Atf3 | Advanced Cell Diagnostics | ACD:426891 | |
| Commercial assay/kit- RNAscope | RNAscope Probe-Mm-Serpina3n; Serpina3n | Advanced Cell Diagnostics | ACD:430191 | |
| Commercial assay/kit- RNAscope | RNAscope Probe-Mm-Csf1; Csf1 | Advanced Cell Diagnostics | ACD:315621 | |
| Commercial assay/kit- RNAscope | RNAscope Probe-Mm-Tubb3; Tubb3 | Advanced Cell Diagnostics | ACD:423391 | |
| Commercial assay/kit- RNAscope | RNAscope Probe-Mm-Gal; Galanin; Gal | Advanced Cell Diagnostics | ACD:400961 | |
| Commercial Assay/kit- RNAscope | RNAscope Probe-Mm-Npy; Npy | Advanced Cell Diagnostics | ACD:313321 | |
| Commercial assay/kit- RNAscope | RNAscope Probe-Mm-Cckbr; Cckbr | Advanced Cell Diagnostics | ACD:439121 | |
| Commercial assay/kit- RNAscope | RNAscope Probe-Mm-Ecel1; Ecel1 | Advanced Cell Diagnostics | ACD:475331 | |
| Commercial assay/kit- RNAscope | RNAscope Probe-Mm-Gpr151; Gpr151 | Advanced Cell Diagnostics | ACD:317321 | |
| Commercial assay/kit- RNAscope | RNAscope Probe-Mm-Nts; Nts | Advanced Cell Diagnostics | ACD:420441 | |
| Commercial assay/kit- RNAscope | RNAscope Probe-Mm-Sox11; Sox11 | Advanced Cell Diagnostics | ACD:440811 | |
| Commercial assay/kit- RNAscope | RNAscope Probe-Mm-Sprr1a; Sprr1a | Advanced Cell Diagnostics | ACD:426871 | |
| Antibody | anti-phospho-c-Jun S63 (rabbit polyclonal) | Cell Signaling | Cell Signaling:9261; RRID:AB_2130162 | 1:500 |
| Antibody | anti-NeuN (mouse monoclonal) | Millipore | Millipore:MAB377; RRID:AB_2298772 | 1:300 |
| Antibody | anti-ATF3 (rabbit polyclonal) | Santa Cruz | Santa Cruz:sc-188; RRID:AB_2258513 | 1:3000 |
| Antibody | anti-Iba1 (rabbit polyclonal) | Wako | Wako:019–19741; RRID:AB_2665520 | 1:200; 1:600 |
| Antibody | Donkey anti-rabbit AlexaFluor 647 | ThermoFisher | ThermoFisher:A31573; RRID:AB_2536183 | 1:500 for immunostaining; 1:600 for iDisco |
| Antibody | Donkey anti-rabbit AlexaFluor 594 | ThermoFisher | ThermoFisher:A21207; RRID:AB_141637 | 1:500 for immunostaining; 1:600 for iDisco |
| Antibody | Donkey anti-mouse AlexaFluor 488 | ThermoFisher | ThermoFisher:A21202; RRID:AB_141607 | 1:500 for immunostaining; 1:600 for iDisco |
| Strain, strain background (mus musculus) | Inbred strain C57BL/6J (mus musculus) | The Jackson Laboratory | Jax:000664; RRID:IMSR_JAX:000664 | |
| Strain, strain background (mus musculus) | STOCK Tg(CAG-cre/Esr1*) 5Amc/J, C57BL/6J (mus musculus) | The Jackson Laboratory | Jax:004453; RRID:IMSR_JAX:004453 | |
| Strain, strain background (mus musculus) | *Map3k12* flox mouse, C57BL/6J (mus musculus) | Genentech, by MTA; PMID 24166713 | | |
| Chemical compound, drug | small molecule GNE-3511 (DLK inhibitor) | Genentech, by MTA; PMID 25341110 | | |
| Software, algorithm | FIJI | NIH | RRID:SCR_002285 | |
| Software, algorithm | arivis Vision 4D | Arivis | | |

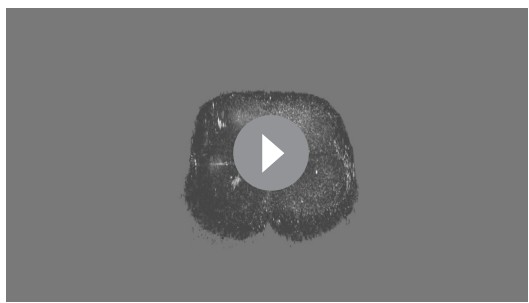

**Video 2.** GNE-3511-treated spinal cord, 5 dpi
DOI: https://doi.org/10.7554/eLife.33910.030

## Mice

All animal care and experimental procedures were performed in accordance with protocols approved by the National Institutes of Neurological Diseases and Stroke and the National Institute of Child Health and Human Development Animal Care and Use Committees. *Map3k12*fl/fl mice (*Le Pichon et al., 2017*; *Pozniak et al., 2013*) were bred to a CAG-CreER transgenic line (Jax#004453). *Map3k12*fl/fl; CAG-CreER−/− mice were referred to as DLK WT; *Map3k12*fl/fl. CAG-CreER+/− mice were referred to as DLK cKO. C57BL/6J mice (Jax#000664) were used in the DLK inhibitor study. Similar numbers of male and female mice were used in all experiments except where stated.

## Tamoxifen administration

*Map3k12*fl/fl; CAG-CreER mice were placed on a diet containing 40 mg/kg Tamoxifen (Envigo TD.130860) for 3 weeks, and back onto regular chow for at least 3 weeks prior to any further testing or tissue harvest. Mice in *Figure 1—figure supplement 1* and *2*, received five consecutive daily doses of 200 mg/kg tamoxifen in corn oil by oral gavage, and received no further treatment for 1 week before receiving SNI. All mice, including Cre negative (CreER−/−) mice, were Tamoxifen-treated.

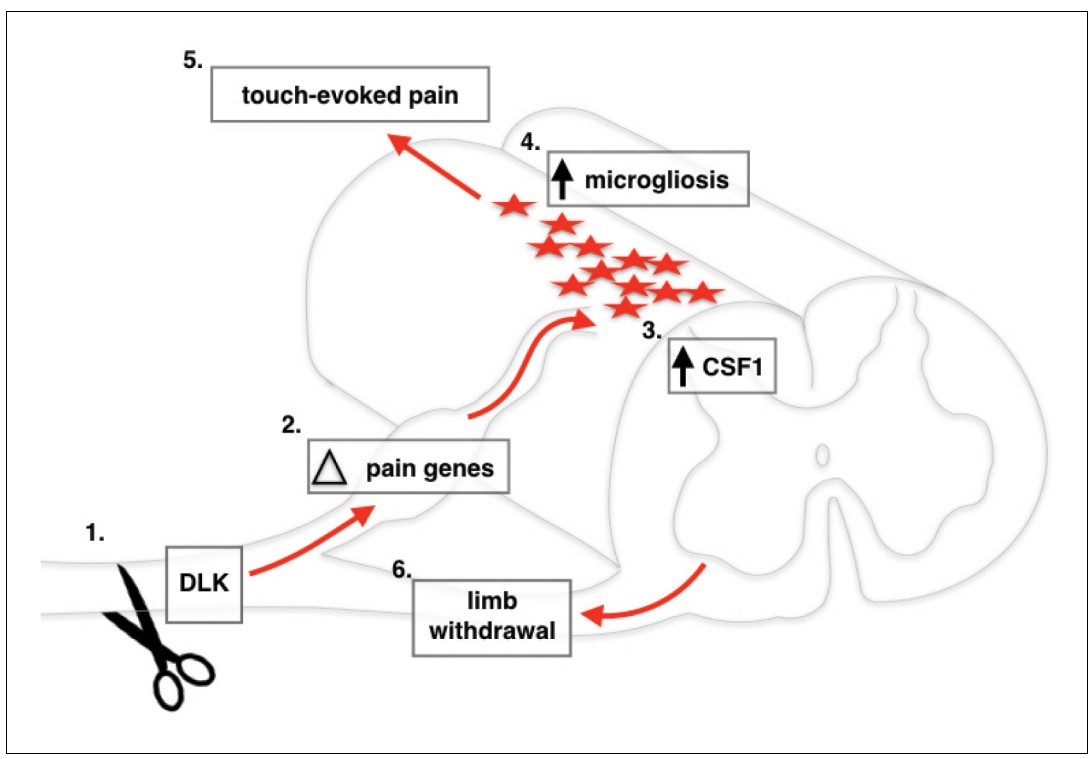

**Figure 6.** Schematic of DLK activation in DRG neurons following spared nerve injury. Nerve injury activates DLK (*Ji and Strichartz, 2004*) which induces a transcriptional response in DRG neurons (*Basbaum et al., 2009*), including Csf1 upregulation (*von Hehn et al., 2012*) which leads to microgliosis in the spinal cord (*Decosterd and Woolf, 2000*). Overall, the transcriptional changes induced by DLK lead to the establishment of touch-evoked pain (*Bourquin et al., 2006*) as evidenced by the paw withdrawal reflex (*Svensson et al., 1993*).
DOI: https://doi.org/10.7554/eLife.33910.031

## Nerve injury models

For intraplantar formalin injections, a 10 µl volume of 5% formalin in PBS was injected into one hind-paw of mice under isoflurane anesthesia. n = 2 male DLK WT, 2 male and one female DLK cKO. For spared nerve injury surgery (SNI), the sural and superficial peroneal branches of the sciatic nerve were ligated (size 8.0 suture thread) and transected, leaving the tibial nerve intact. The muscle tissue overlying the injury was gently placed back together and the skin held together with wound clips which were removed 5 days later. In *Figure 1*, n = 9,DLK WT (4 males and 5 females) and 11 DLK cKO (6 males and 5 females). For sciatic nerve transection, the entire sciatic nerve was transected at the mid-thigh level. The overlying muscle was placed back together and the skin was held together with wound clips. For experiments in *Figures 1–3*, *5D-H*, no post-operative analgesia was provided. For experiments in *Figures 4*, *5A–C*, and in *Figure 1—figure supplement 1*, the following analgesia was provided on the day of surgery and for the next 2 days: buprenorphine (0.05 mg/kg) once daily IP for 3 days, and ketoprofen (5 mg/kg) once daily subcutaneously for 3 days.

## Von Frey behavior test (static allodynia)

Behavioral experiments were done blind to genotype or treatment. Mice were habituated for 15–30 min to inverted glass staining jars (10 cm long x 8.5 cm wide x 7 cm tall) placed on a wire mesh platform. White paper was placed between each chamber so the mice could not see each other. Only mice of the same sex were tested in the same session. Von Frey filaments were manually applied to the center of the mouse's ipsilateral hind paw (territory innervated by the tibial nerve). The following filaments were tested: 0.008, 0.02, 0.04, 0.07, 0.16, 0.4, 0.6, 1, 1.4 g. Testing was performed essentially as described in (*Latremoliere et al., 2015*). Each animal received 10 stimulations with each filament. The inter trial interval was at least 15 s. If a mouse showed paw withdrawal responses or escape attempts for five trials or more out of 10, that filament force was considered its mechanical threshold. Testing began using the 0.04 g filament. If a mouse responded to a given filament, the following stimulus used the next lowest filament. If a mouse failed to show a withdrawal response, the following stimulus used the next highest filament. Once a mouse responded all 10 times to a given filament, no further testing of higher force filaments was performed, but they were scored as a 10/10 for graphing and analysis purposes.

## Light brushing behavior test (dynamic allodynia)

Mice were lightly stroked on the base of their injured paw from their heel towards the center of their paw pad with a trimmed paintbrush (5/0) and at a velocity of ~2 cm/s as described in (*Cheng et al., 2017*). Scoring system: 0 – rapid paw lift (typical response of an uninjured mouse; lasts <1 s); 1 – sustained paw lift into the body (>2 s); 2 – lateral kick or startle-like jump; 3 – paw licking (or tending to paw with mouth) or multiple flinching responses. Each trial consisted of up to five brush strokes, where if the mouse responded with a score greater than 0, the trial was terminated with that score. The inter-trial interval was at least 3 min. Each mouse was given four trials and the average score was calculated.

## GNE-3511 dosing

GNE-3511 was synthesized as described in (*Patel et al., 2015*) and formulated in a solution of 0.5% w/v USP Grade Methyl Cellulose and 0.2% v/v Tween 80 in water. The solution was sonicated and stored at 4°C for no more than 7 days. Mice were dosed twice daily (BID) at 75 mg/kg with a 10 ml/kg of a 7.5 mg/ml solution of GNE-3511 or vehicle alone by oral gavage. The last dose was administered 2 hr prior to tissue harvest.

## Tissue harvest and preparation

Mice were anesthetized with isoflurane, decapitated. L3 and L4 DRGs were harvested from the contralateral and ipsilateral sides to the injury, and snap frozen on dry ice. The DRGs were embedded in OCT and sectioned at 10–20 µm for staining or in situ hybridization. For spinal cord harvest, mice were transcardially perfused with PBS and decapitated. A cut was made through the spinal column at the level of the hip bones (sacral) and a PBS-filled syringe fitted with a 20G needle (7.25 mm) was inserted into the caudal end of the spinal column to flush the spinal cord out. Spinal cords were then fixed in 4% paraformaldehyde overnight at 4°C.

## Immunostaining

Fresh frozen sections were briefly fixed in 4% paraformaldehyde in phosphate buffered saline (PBS) for 15 min on ice, then washed in PBS followed by PBS containing 0.1% Triton X (PBSTx). Sections were blocked in 5% normal donkey serum in PBS for 1–2 hr at room temperature and incubated with primary antibody diluted in 0.5% normal donkey serum in PBS overnight at 4°C. Slides were washed in PBSTx, then incubated in AlexaFluor-conjugated secondary antibodies (ThermoFisher) in PBS for 2 hr at room temperature, followed by two washes in PBSTx and two in PBS. Slides were coverslipped using Prolong Gold Antifade Mountant containing DAPI (4',6-diamidino-2-phenylindole) nuclear counterstain (ThermoFisher #P-36931).

## Antibodies

Primary antibodies: rabbit anti-ATF3 (Santa Cruz #sc-188, 1:3000), rabbit anti-phospho-c-Jun (Cell Signaling #9261, 1:500), mouse anti-NeuN (Millipore #MAB377, 1:300).

Secondary antibodies: donkey anti-mouse and donkey anti-rabbit AlexaFluor conjugates (ThermoFisher; 1:500).

## RNAscope in situ hybridization

Multiplex in situ hybridization was performed according to the manufacturer's instructions (Advanced Cell Diagnostics #320850). Briefly, 10–16 µm-thick fresh frozen sections were fixed in 4% paraformaldehyde, treated with Proteinase K , and hybridized with gene-specific probes to mouse *Atf3, Cckbr, Csf1, Ecel1, Gal, Gpr151, Npy, Nts, Serpina3n, Sox11,* and *Sprr1a. In situ* experiments shown in *Figures 2* and *3* were from L3 (lumbar level 3) DRGs. All other in situ data used lumbar 4 (L4) DRGs.

## iDisco procedure and imaging

The iDisco protocol was followed as described in (*Renier et al., 2014*) and https://idisco.info. Rabbit anti-Iba1 (Wako #019–19741) was used at 1:100 in *Figure 3* at 1:200 in *Figure 5* during a 3 day incubation at 37°C. Secondary antibody AlexaFluor594-conjugated donkey anti-rabbit (ThermoFisher #A21207) was used at 1:200 in *Figure 3*, and an AlexaFluor647-conjugate at 1:600 in *Figure 5*. Spinal cords were embedded in 1% agarose prior to clearing to facilitate handling for imaging. Imaging of iDisco samples was done using a two-photon microscope (Olympus MPE-RS; representative images in *Figures 3* and *5*) as well as a light sheet microscope (LaVision Biotec Ultramicroscope II; quantification of all spinal cord samples in *Figures 3* and *5*, and videos). Imaging on the LaVision Biotec UltraMicroscope II was performed using their 4x objective. All images were acquired using six light-sheets with dynamic focusing (16 positions) enabled. For the Dynamic Focus Processing and Merge Lightsheet calculations, the Contrast Adaptive and Contrast algorithms were respectively used.

## Imaging and analysis of immunostaining and in situ hybridization in Figures 1–3

Sections were imaged by confocal microscopy (Olympus FV1000; Zeiss LSM780) and analyzed using FIJI software (National Institutes of Health) by an experimenter blind to genotype or treatment. For immunostaining in DRG, ATF3, p-c-Jun, and NeuN positive-stained cells were counted manually in 3 images per animal. For gene expression by in situ hybridization in DRG in *Figures 2*, *3*, 4-6 DRG sections from each animal were analyzed and averaged, with n = 3 male mice per genotype. ROIs were drawn around the areas of DRG containing cell bodies, excluding areas containing only nerve. Images were thresholded into a binary signal and the percent area of fluorescent signal was quantified.

## Imaging and analysis of gene expression by in situ hybridization in Figures 4–5

Confocal stacks were acquired on a Zeiss LSM780 or 800 using a 20X objective. Laser power, gain, and digital offset were kept constant for all channels other than DAPI. Tile region and z-stack range varied for individual DRGs, with a range of 8–20 z-stacks depending on tissue organization. RNA expression intensities were obtained and quantified using an in-house built segmentation pipeline in

Arivis Vision4D. For all images, the range of the *Tubb3*-positive region over multiple z-stacks was used to define the working volume. This region was isolated from surrounding nerve tissue using a denoising filter followed by a blob finder. A curvature flow filter with 20 iterations and a working time of 0.1 was used to denoise the *Tubb3* signal. Next a blob finder with a diameter of 50 µm, threshold of 1.00 and split sensitivity of 96 was applied to the denoised image stack in order to define the *Tubb3* region into multiple small volumes. (Images taken on the ZEISS MIC800 used a blob finder with a diameter of 40 µm, threshold of 1.00 and split sensitivity of 93.4 to achieve similar results). Blob finder volumes that did not correspond with DAPI positive regions of tissue were deleted manually. Remaining volumes were then summed to produce a total *Tubb3*-positive volume specific to each DRG. RNA probe intensities within these *Tubb3* regions were summed to give the total RNA intensity per *Tubb3* volume for each individual DRG.

### Analysis of Iba1 signal in spinal cord

For Iba1 signal shown in *Figures 3* and *5*, 4–6 images from Z planes regularly spaced within the entire dorsal horn region of the spinal cord were analyzed and averaged per animal. ROIs were manually drawn, images thresholded, and the percent area of Iba1-positive signal quantified. For quantification of Iba1-positive volumes or 'clouds' in *Figure 4*, the following method was employed. Tiff images acquired on the Ultramicroscope II were converted into .SIS files. In arivis Vision4D (2.12.6), an in-house developed analysis pipeline was run on each sample. The pipeline created a 3D annotation to capture the size and intensity of regions containing many Iba1+ cells. This was done by the use of a rigorous De-Noising Mean Filter with a voxel radius of 10, after which a 3D Intensity Threshold was applied, and to which a Segment Filter of 0.001 mm$^3$ was used to filter out very small annotations.

### Videos

Image stacks taken with the Ultramicroscope II were converted to 3D video files using arivis software (Vision 4D).

### Statistical analysis

All data were analyzed using Graphpad Prism software. Specific tests employed are described in each figure legend.

## Acknowledgements

DLK$^{lox/lox}$; CAG-Cre$^{ERT2}$ mice and DLK inhibitor GNE-3511 were obtained from Genentech by MTA. We would like to thank the NINDS animal care technicians and veterinary staff for assistance with mouse care, as well as Nick Ryba, Mark Hoon, Joseph Lewcock, and Morgan Sheng for helpful discussions and comments on the manuscript. This work was funded by the Intramural Program at National Institutes of Health (IRP-NIH) through NCCIH (ATC) and NICHD (CELP). Further support was provided by the NIH DDIR Innovation Award (CELP and ATC).

## Additional information

### Funding

| Funder | Grant reference number | Author |
| --- | --- | --- |
| National Center for Complementary and Integrative Health | Intramural Research Program | Alexander T Chesler |
| National Institutes of Health | Intramural Research Program - DDIR Innovation Award | Alexander T Chesler Claire E Le Pichon |
| National Institute of Child Heath and Human Development | Intramural Research Program | Claire E Le Pichon |

The funders had no role in study design, data collection and interpretation, or the decision to submit the work for publication.

## Author contributions
Josette J Wlaschin, Jacob M Gluski, Data curation, Formal analysis, Investigation, Visualization, Methodology, Writing—review and editing; Eileen Nguyen, Data curation, Formal analysis, Investigation, Methodology; Hanna Silberberg, Data curation, Formal analysis, Investigation, Methodology, Writing—review and editing; James H Thompson, Investigation; Alexander T Chesler, Claire E Le Pichon, Conceptualization, Resources, Data curation, Formal analysis, Supervision, Funding acquisition, Investigation, Visualization, Methodology, Writing—original draft, Project administration, Writing—review and editing

## Author ORCIDs
Alexander T Chesler  https://orcid.org/0000-0002-3131-0728
Claire E Le Pichon  http://orcid.org/0000-0002-9274-3615

## Ethics
Animal experimentation: All animal care and experimental procedures were performed in accordance with protocols approved by the National Institutes of Neurological Diseases and Stroke Animal Care and Use Committee (protocols #1365-14, 1369-14, and 1365-17) and by the National Institute of Child Health and Human Development Animal Care and Use Committee (protocol #17-003).

## Decision letter and Author response
Decision letter https://doi.org/10.7554/eLife.33910.036
Author response https://doi.org/10.7554/eLife.33910.037

# Additional files

## Supplementary files
• Transparent reporting form
DOI: https://doi.org/10.7554/eLife.33910.032

## Data availability
All data generated or analysed during this study are included in the manuscript and supporting files. Source data files have been provided for Figures 1-5.

The following previously published dataset was used:

| Author(s) | Year | Dataset title | Dataset URL | Database, license, and accessibility information |
| --- | --- | --- | --- | --- |
| Guan Z, Kuhn JA, Wang X, Colquitt B, Solorzano C, Vaman S, Guan AK, Evans-Reinsch Z, Brax J, Devot M, Abboud-Werner SL, Lanier LL, Lomvardas S, Basbaum AI | 2016 | Injured sensory neuron-derived CSF1 induces microglial proliferation and DAP12-dependent pain. | https://www.ncbi.nlm.nih.gov/query/acc.cgi?acc=GSM1626431 | Publicly available at the NCBI Gene Expression Omnibus (accession no: GSM1626431 from series GSE66619). |

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
