## [Decision Letter]

Thank you for submitting your article "Dual Leucine Zipper Kinase is required for mechanical allodynia and microgliosis after nerve injury" for consideration by *eLife*. Your article has been favorably evaluated by a Senior Editor and two reviewers, one of whom is a member of our Board of Reviewing Editors. Your article has been reviewed by 2 peer reviewers, and the evaluation has been overseen by a Reviewing Editor and a Senior Editor. The reviewers have opted to remain anonymous.

The reviewers have discussed the reviews with one another and the Reviewing Editor has drafted this decision to help you prepare a revised submission.

Summary:

The manuscript by Le Pichon et al. demonstrates an important role for the dual leucine zipper kinase (DLK) in static mechanical allodynia. Authors demonstrate that mice with a deletion of the kinase in the adult show significantly reduced static allodynia with no effect on baseline von Frey threshold. Systemic delivery of an inhibitor of DLK twice per day for a week also reduced the allodynia measured at one week. The authors also show that microglia in the dorsal horn (detected with Iba1 staining) after SNI are reduced in the absence of DLK or with the inhibitor. Next the authors demonstrate (in the supplementary) that induction of ATF3 and p-c-Jun by SNI or formalin injection are reduced in DLK cKO or with the inhibitor. Based on this, they look at the change in mRNA levels for genes thought to be downstream of these proteins (*Gal, NPY, Csf1, Serpina3n*) in the cKO after SNI, demonstrating changes consistent with their proposed roles at least for the last two genes. Finally, the authors use a database of genes that were shown to be upregulated by SNI, they pick 7 of these and then show that they are not upregulated in the cKO after SNI (also in supplementary).

The paper is following on the heels of work by this author and others that DLK has a role in the stress response to nerve damage and that inhibiting it can be beneficial after the insult. In earlier work, Pichon and previous colleagues performed a transcriptome study of the genes that are up and down regulated 3 days after optic nerve crush in inhibitor and vehicle treated animals. Induction of ATF3 was identified as being suppressed by the inhibitor. A decrease in the amount of p-c-Jun was also demonstrated in this study.

The data are solid and the authors do a nice job of suggesting a potential link between DNK and the regulation of genes previously shown to be important for nerve injury induced mechanical allodynia induced with *Csf1* and *Serpina3n* being the main ones.

Essential revisions:

1) Although the simplest interpretation of the authors' findings is that DLK is required cell-autonomously in DRG sensory neurons, it is possible that the DLK gene is required in some cell type other than primary sensory neurons because the CreER line used (ROSA-CAG-CreERT2) is ubiquitously expressed. This issue needs to be addressed with experiments, if possible, or if not then it should be discussed in detail.

2) What is the earliest time-point that one can observe punctate allodynia after SNI? Also, what is the rationale for delivering the inhibitor for a week before testing the allodynia? Some discussion of this should be made. Allodynia measured at 16 hours, and daily for a week would be better.

3) A strength of the study is the potential for translating the findings for treatment of chronic pain after nerve injury. The reviewers would like to see additional experiments that ask how long after SNI surgery can the drug be applied and still show an effect? A later time point after injury (e.g. 3-7 days) to test whether it still has an effect would be particularly informative since most patients seek relief later and this drug has been implicated in reversing degeneration and nerve damage.

4) What is the evidence for efficacy and specificity of GNE-3511 towards DLK in DRG neurons in vivo? If there is a measure of DLK activity, does GNE-3511 lead to inhibition of this activity in DRG neurons at doses that correspond to those needed for inhibition of mechanical allodynia and gliosis?

---

## [Author Response]

Essential revisions:1) Although the simplest interpretation of the authors' findings is that DLK is required cell-autonomously in DRG sensory neurons, it is possible that the DLK gene is required in some cell type other than primary sensory neurons because the CreER line used (ROSA-CAG-CreERT2) is ubiquitously expressed. This issue needs to be addressed with experiments, if possible, or if not then it should be discussed in detail.

As requested we now include a detailed discussion on this point in the manuscript (Discussion, second paragraph).

We agree it would have been ideal to generate a mouse with sensory neuron-specific deletion of DLK, but the timeline to generate such mice was impractical.

2) What is the earliest time-point that one can observe punctate allodynia after SNI? Some discussion of this should be made. Allodynia measured at 16 hours, and daily for a week would be better.

We have addressed this point in a new experiment to see how early punctate allodynia can be observed. We tested DLKWT vs. DLKcKO mice at 1, 3, 5 and 7 days post injury (dpi) (Figure 1—figure supplement 2). At 1 dpi, control mice (but not the knockouts) displayed mechanical allodynia; however, sensitization was far lower at 3 and 5 days post-SNI. We have concerns that the mice are still recovering from surgery e.g. damage to muscle and skin (and presence of wound clips) as well as nerve injury during this first week. Reliable allodynia is observed in the control group at 7 dpi. See text in Figure 1—figure supplement 2 legend.

Notably, we also measured the time-course for transcriptional changes and microgliosis and we show that these occur over the first week after nerve injury in parallel with the development of static mechanical allodynia. These new data are shown in Figure 4.

Also, what is the rationale for delivering the inhibitor for a week before testing the allodynia?

We now discuss the rationale for the inhibitor experiment design in the subsection “Treatment with DLK inhibitor GNE-3511 prevents injury-induced transcriptional changes, mechanical allodynia and microgliosis”, based on results shown in Figure 4.

3) A strength of the study is the potential for translating the findings for treatment of chronic pain after nerve injury. The reviewers would like to see additional experiments that ask how long after SNI surgery can the drug be applied and still show an effect? A later time point after injury (e.g. 3-7 days) to test whether it still has an effect would be particularly informative since most patients seek relief later and this drug has been implicated in reversing degeneration and nerve damage.

The focus of our study is the role for DLK in establishing pain sensitivity. We agree it would be interesting to test the effect of starting DLK inhibitor treatment at different time points after injury. But because of the competing effects of SNI and blocking DLK this would really require a comprehensive evaluation (rather than a single time point) and is beyond the scope of the present study. Importantly, our new data do demonstrate that blocking DLK reverses *Atf3* induction: compare normal *Atf3* induction at 18h (Figure 4) with 5 dpi (Figure 5); discussed in the subsection “Treatment with DLK inhibitor GNE-3511 prevents injury-induced transcriptional changes, mechanical allodynia and microgliosis”.

4) What is the evidence for efficacy and specificity of GNE-3511 towards DLK in DRG neurons in vivo? If there is a measure of DLK activity, does GNE-3511 lead to inhibition of this activity in DRG neurons at doses that correspond to those needed for inhibition of mechanical allodynia and gliosis?

We have previously generated data on the pharmacokinetic and pharmacodynamic properties of GNE-3511 towards DLK, as well as its efficacy and specificity (Patel et al., 2015, Le Pichon et al., 2017). This guided our selection of the dose level and dosing frequency. In our published work, we have shown that GNE-3511 has the expected pharmacodynamic effect for DLK inhibition in vivo by reducing phospho-c-Jun in retinal ganglion cells and motor neurons. Here, in our revision, we have extended this to DRG neurons as we show in Figure 5A-C the efficacy of GNE-3511 treatment on downstream transcriptional targets of DLK. GNE-3511 prevents the upregulation of *Atf3* and *Csf1* in DRG neurons at 5 days post SNI, just as we saw in the DLK cKO (shown in Figure 2).